# Barriers against Implementation of European Society of Gastrointestinal Endoscopy Performance Measures for Colonoscopy in Clinical Practice

**DOI:** 10.3390/medicina60071166

**Published:** 2024-07-19

**Authors:** Giulia Gibiino, Leonardo Frazzoni, Andrea Anderloni, Lorenzo Fuccio, Alessandro Lacchini, Cristiano Spada, Carlo Fabbri

**Affiliations:** 1Gastroenterology and Digestive Endoscopy Unit, Forlì-Cesena Hospitals, Ausl Romagna, 47121 Forlì, Italy; giulia.gibiino@gmail.com (G.G.); leonardo.frazzoni@gmail.com (L.F.); carlo.fabbri@auslromagna.it (C.F.); 2Gastroenterology and Endoscopy Unit, Fondazione IRCCS Policlinico San Matteo, 27100 Pavia, Italy; 3Gastroenterology Unit, Department of Medical and Surgical Sciences, IRCCS Azienda Ospedaliero-Universitaria di Bologna, University of Bologna, 40100 Bologna, Italy; lorenzofuccio@gmail.com; 4LML Avvocati Associati, 20121 Milano, Italy; avv.lacchini@lmlassociati.it; 5Digestive Endoscopy Unit, Fondazione Policlinico Universitario A. Gemelli IRCSS, 00042 Rome, Italy; cristiano.spada@policlinicogemelli.it

**Keywords:** electronic reporting systems, quality indicators, high-quality endoscopy, adenoma detection rate, high-quality colonoscopy

## Abstract

*Background and Objectives:* The implementation and monitoring of the European Society of Gastrointestinal Endoscopy (ESGE) performance measures for colonoscopy are suboptimal in clinical practice. Electronic reporting systems may play an important role in data retrieval. We aimed to define the possibility of systematically assessing and monitoring ESGE performance measures for colonoscopy through reporting systems. *Materials and Methods*: We conducted a survey during a nationwide event on the quality of colonoscopy held in Rome, Italy, in March 2023 by a self-administered questionnaire. Analyses were conducted overall and by workplace setting. *Results*: The attendance was 93% (M/F 67/26), with equal distribution of age groups, regions and public or private practices. Only about one-third (34%) and 21.5% of participants stated that their reporting system allows them to retrieve all the ESGE performance measures, overall and as automatic retrieval, respectively. Only 66.7% and 10.7% of respondents can systematically report the cecal intubation and the adenoma detection rate, respectively. The analysis according to hospital setting revealed no significant difference for all the items. *Conclusions*: We found a generalized lack of systematic tracking of performance measures for colonoscopy due to underperforming reporting systems. Our results underline the need to update reporting systems to monitor the quality of endoscopy practice in Italy.

## 1. Introduction

Colorectal cancer (CRC) is currently the third most commonly diagnosed neoplasia and the second-leading cause of cancer deaths worldwide. Current expectations show an increase in the coming years, particularly in young adults, in relation to dietary habits in Western countries and the rising obesity rate [1,2]. The spread of screening programs nationwide has allowed for a heterogeneous variation in incidence across Italy [3,4]. However, 48,000 new diagnoses were registered in 2022, and this disease still accounts for high direct and indirect costs within the healthcare systems. The diagnostic and therapeutic impact of digestive endoscopy is strongly influenced by the quality of the endoscopic examination. Over the years, the European Society of Digestive Endoscopy (ESGE) has identified quality measures to be satisfied for lower gastrointestinal (GI) endoscopy [5,6,7]. Most of these are performance measures, nevertheless with a proven association with clinical outcomes [8]. For example, the adenoma detection rate (ADR) is an established surrogate measure which is inversely related to missed lesions and CRC incidence after colonoscopy [8]; withdrawal time and quality of bowel preparation are related to adenoma detection and CRC prevention as well. Further, the appropriateness of the indication significantly increases the diagnostic yield of colonoscopy in terms of CRC and relevant findings [9]. However, the implementation of these performance measures in clinical practice is still suboptimal [10]. The insufficient awareness of physicians of the clinical relevance of these measures has been traditionally considered as the main reason behind such suboptimal implementation [5,10,11,12,13]. However, the inadequate development of electronic reporting systems may play an important role, as reporting systems should facilitate easy data retrieval at any time in a universally compatible format [14].

The aim of this survey was to define, in a sample of Italian endoscopists, the degree of knowledge of ESGE performance measures and the actual possibility of systematically assessing and tracking ESGE performance measures through electronic reporting systems.

## 2. Materials and Methods

This survey was conducted during a nationwide event on the quality of colonoscopy held in Rome, Italy, 10–11 March 2023. All participants were gastroenterologists performing endoscopy, excluding those still in training or retired. A self-administered questionnaire was provided to and returned by participants during the event. The questionnaire can be consulted in the Appendix A.

There were no fees or incentives for participation in the survey. Written informed consent for the anonymous use of data provided in the questionnaire was individually obtained from all participants.

### 2.1. Questionnaire

The questionnaire was developed by a scientific committee composed of members of the event faculty (CF, GG, LFr, LFu). The committee selected a set of questions, taking into consideration the demographic and professional characteristics of participants and their awareness of—and the possibility of monitoring owing to electronic reporting systems—ESGE guidelines and performance measures for lower GI endoscopy in clinical practice. In detail, questions were selected according to ESGE performance measures for lower GI endoscopy published in 2017 [6], ESGE guidelines on bowel preparation [15] and ESGE guidelines on endoscopy reporting systems [10]. The questionnaire consisted of three sections including a total of 38 multiple-choice questions. The majority of questions were formulated as yes/no; for some answers, categories were considered, e.g., <5%, 5–15%, 10–25%, 25–50%, >50%. The complete original questionnaire is available in the Appendix A section.

The first section contained 8 questions regarding the demographic and professional characteristics of participants.

The second section contained 4 questions regarding cultural aspects and 21 questions about pre-procedure assessment. The key measures assessed in this section were routinary audits for endoscopists’ quality; knowledge and updates about performance measures recommended by the ESGE and Italian Society of Digestive Endoscopy (SIED); clinical impact considered for performance measures. Pre-procedure assessment included indication and appropriate colonoscopy; registration and reporting about appropriate colonoscopy; bowel preparation; adenoma detection rate (ADR); standardized reporting about bowel preparation, re-preparation and resected lesions.

The third section included post-procedure measures: reporting about patients’ satisfaction after bowel preparation and overall experience; reporting about delayed post-colonoscopy complications; a structured system for indications to follow-up colonoscopies. 

### 2.2. Statistical Analysis

A descriptive analysis was conducted. Continuous variables were reported as means and standard deviations (SDs) if normally distributed, or otherwise, as medians and interquartile ranges. Categorical variables were described as absolute proportions and percentages. The chi-square test or Fisher’s exact test were applied as appropriate to assess significant correlations between categorical variables. Multivariable logistic regression analyses were performed to identify factors related to structured reporting systems for colonoscopy.

A *p*-value < 0.05 was considered statistically significant. Statistical analysis was performed using STATA 16 (Stata Corp., College Station, TX, USA).

## 3. Results

Among 100 participants, 93 completed the survey; 67 (72.04%) were male, with similar distribution among different age groups, ranging from 30 to 70 years. Participants’ characteristics are reported in Table 1. Figure 1 shows the distribution of participants’ workplaces around Italy. Most participants (74.2%) worked in centers with accredited activity for CRC screening. The majority of attendees was represented by heads of endoscopy units or doctors, representing, in any case, medium- or high-volume centers.

Nearly all (n = 89, 97.8%) participants confirmed knowledge of ESGE performance measures and awareness of their clinical impact. Only about one-third (n = 31, 34%) of participants stated they can retrieve all the ESGE performance measures from their reporting systems; further, automatic retrieval was reported in 20 (21.5%) cases only. Indication to colonoscopy was stated in a structured way in the minority of cases (n = 13, 14.1%). Concerning bowel preparation, the assessment of adequate cleansing, the adoption of a validated cleansing scale, the type of preparation, the modality of intake and the time gap before the beginning of the exam were explicitly assessed by reporting systems in 28 (30.4%), 61 (65.6%), 26 (27.9%), 20 (21.7%) and 8 (8.6%) cases, respectively.

More than half (n = 62, 66.7%) of participants reported that their reporting system assessed the completeness of colonoscopy, whereas only a minority of participants (n = 10, 10.7%) stated that their reporting system allowed them to systematically assess their ADR. Less than half (n = 39, 42.4%) of endoscopists were allowed to assess the appropriate polypectomy technique, and almost none of the reporting systems monitored patient satisfaction about bowel preparation (6.4%) and colonoscopy experience (4.3%). About half of participants stated that they are allowed by their reporting system to systematically monitor complications (n = 44, 47.3%) and to provide a follow-up indication after the examination (n = 52, 55.9%).

The analysis according to workplace setting revealed no significant difference for all the considered items, yielding overall low figures for the retrieval of all ESGE quality indicators, being 18.7%, 37.7% and 36.3% for academic centers, community hospitals and private facilities, respectively. Also, the ability to systematically monitor the ADR based on the reporting system was as low as 6.3%, 13.2% and 8.3% among academic centers, community hospitals and private facilities, respectively. Details can be found in Table 2.

On multivariate analysis adjusted for the age and sex of the respondent, the only variable significantly associated with reporting systems allowing to assess and monitor ESGE performance measures for colonoscopy was a reporting system set up within the last five years (OR 5.51, 95% CI). Univariate and multivariate analyses are shown in Table 3.

## 4. Discussion

In the present survey conducted among 100 Italian endoscopists from all over Italy, we found that most of the participants were aware of ESGE performance measures for colonoscopy, whereas most of them were not allowed to systematically assess and monitor such measures by electronic reporting systems. Such a finding was confirmed irrespective of workplace setting when considering academic centers, community hospitals and private facilities. The main explanation for this suboptimal result at multivariate analysis turned out to be that most centers have out-of-date reporting systems. Academic hospitals were the least represented in the sample, but this point must be assessed in view of the overlap that is often observed in Italy between community centers and university locations. In other words, we wanted to define whether the presence of a training course has so far led to better investments in terms of technology in reporting systems. Although the training of young endoscopists takes place there on a daily basis, the academic locations are also burdened by the same problems as the other institutions.

Quality improvement aims at a better management of our patients and a better use of our resources. Knowledge of quality indicators is essential but should be necessarily accompanied by an electronic reporting system with adequate tracking of all key variables and standardization [6,10]. This allows for effective measurement, comparison and possible improvements also on a national scale. As far as various European-wide surveys have shown so far, the main obstacles to the implementation of performance measures are resistance to change, lack of regulation and the practicality of measuring performance measures [7].

The suboptimal adherence to performance measures in endoscopy is a known fact in Italy, and our study has a consistent trend [10,11]. In addition, the data of the patient’s reported experience do not emerge as relevant in the current approach and do not follow standardized monitoring [12]. Even more important is the evidence that emerged regarding reporting systems. According to the ESGE [13,14,15,16,17,18], reporting systems should allow automatic data extraction to track performance measures. Our survey showed that this is not feasible in clinical practice in most centers in Italy. Further, academic endoscopists are not allowed to adequately report on ESGE performance measures which have been adopted by the Italian Society of Digestive Endoscopy (SIED). This is crucial, as they host trainees and younger endoscopists in training. In fact, as performance measures were first and foremost suggested as a measure of progress in the various training phases, the systematic assessment and tracking of performance measures might pave the way for educational programs promoted by local and national societies [9].

The absence of reporting systems allowing endoscopists to track their performance measures means they are not permitted to conduct audits and re-training programs to improve the performance of underperforming endoscopists and endoscopy services. Looking at the significant factors in our multivariable analysis, outdated reporting systems did not allow them to monitor ESGE performance measures. It is not unexpected that reporting systems introduced within the last five years significantly relate to structured reporting, probably owing to the advancement of informatics that should be at the service of clinical practice [5]. Considering other countries, the Dutch registry is the most important model proposed [19]. Specifically, the first, the Dutch Gastrointestinal Endoscopy Audit (DGEA), allows a focus on the quality of colonoscopy [20]. Data of all colonoscopies in participating services are automatically extracted from standardized endoscopy reports without additional administrative burden. This registry was introduced in 2013 as an implementation of the Dutch Institute for Clinical Auditing (DICA), and 48 hospitals or endoscopy centers voluntarily participated, with overall missing values limited to <1%. It is a fundamental example that does not change the daily workflow and does not cause additional administrative work, relying on a uniform reporting system and connection with the local endoscopy reporting systems, thereby allowing automated data extraction of the original source data from hospitals or endoscopy centers.

A similar attempt outside Europe, the Quality Improvement and Delivery Science (QUIDS) Program, was introduced in 2014 in Pennsylvania, and it should be considered as another virtuous example of a clinician-led project in order to show that quality measurement is part of everyday practice and not part of abstract concepts [21]. 

The use of established systems that allow for data extraction comparable among different institutions offers several advantages already shown in the recent literature. Belderbos et al. showed their experience in 2015 by comparing quality parameters of routine colonoscopies between two academic and five nonacademic hospitals in The Netherlands. They included 3129 consecutive patients undergoing colonoscopy, and they proposed a colonoscopy quality indicator (CQI) by combining cecal intubation rate (CIR) and ADR [22]. This new tool should lead to an evaluation that is less focused on the performance of the individual endoscopist and rather promotes changes that benefit the entire institution [23]. We consider our analysis to be an original contribution with several strengths, including a high adherence rate over 90%. Further, it is the first survey to investigate the state of the art of reporting systems in Italy and the extent to which they allow the automatic tracking of performance measures in colonoscopy. Other countries are showing how important it is to set up national registers to monitor quality in colonoscopy [24,25]; in our area, the high heterogeneity and reporting methods make a detailed picture difficult, but certainly, the intention of our research was to raise this issue. Our study, as a survey, has many limitations due to its small sample size and data collection based on spontaneous adherence by endoscopists; however, as it is a good representation of the various realities on a national scale, it could be a starting point for various regional alignment and even national initiatives, as well as a comparison with other countries. It is not known to date how many realities adopt an electronic reporting system shared between several centers and how a precise measure of improvement from year to year takes place.

## 5. Conclusions

In conclusion, our study showed a generalized lack of standardized tracking of performance measures for colonoscopy around Italy due to outdated reporting systems. At this stage in Italian practice, we think it is recommended to unequivocally include in reporting systems the main performance measures recommended by the ESGE. Our results underline the need to urgently update reporting systems to monitor the quality of endoscopy practice in Italy and promote improvement programs [26]. Multicenter and large-sample studies are needed to demonstrate the advantage of a uniform reporting system; also, an equal comparison between European countries would allow for measurable progress and a clearer application of guideline recommendations. 

## Figures and Tables

**Figure 1 medicina-60-01166-f001:**
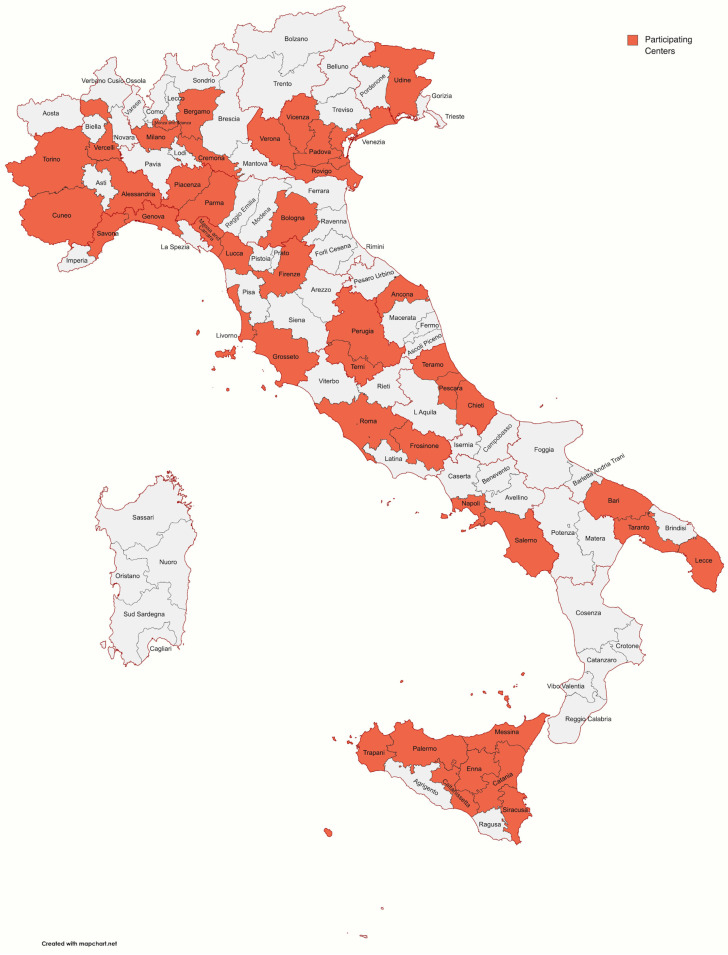
Map showing the distribution of participants’ workplaces around Italy.

**Table 1 medicina-60-01166-t001:** Participants’ characteristics.

Participants N = 93 N (%)	
Gender Male Female	67 (72.04%) 26 (27.96%)
Age, years 30–40 41–50 51–60 61–70	14 (15.05%) 35 (37.63%) 31 (33.33%) 13 (13.98%)
Region North-east North-west Center South and islands	18 (19.78%) 20 (21.98%) 24 (26.37%) 29 (31.8%)
Type of center Academic hospital Community hospital Private hospital Private affiliated to public health system	16 (17.20%) 53 (56.99%) 10 (10.75%) 14 (15.05%)
Colorectal cancer screening center	69 (74.19%)
Age of reporting system, years <5 6–10 11–15 >15	35 (37.63%) 28 (30.11%) 12 (12.90%) 18 (19.35%)
Participation in periodical audit	30 (32.97%)
Attendance of courses about quality in endoscopy	86 (94.51%)
Knowledge of quality indicators	89 (97.80%)

**Table 2 medicina-60-01166-t002:** Reporting systems in terms of assessment and extraction of ESGE performance measures compared between academic, community and private centers.

	Academic Hospital (n = 16)	Community Hospital (n = 53)	Private Hospital/Private Affiliated PHS (n = 24)	*p* Value
Quality indicator promoted by ESGE	3 (18.75%)	20 (37.74%)	8 (36.36%)	0.40
Indication to colonoscopy	2 (13.33%)	8 (15.09%)	3 (12.50%)	1.00
Percentage of adequate bowel preparation	6 (37.50%)	17 (32.69%)	5 (20.83%)	0.47
Compilation of validated bowel preparation scale	11 (68.75%)	35 (66.04%)	15 (62.50%)	0.91
Type of bowel preparation administered	2 (12.50%)	16 (30.19%)	8 (33.33%)	0.33
Bowel preparation modality (i.e., split dose)	2 (13.33%)	13 (24.53)	5(20.83%)	0.68
Time gap between preparation and exam	0	5 (9.43%)	3 (12.50%)	0.45
Complete colonoscopy (i.e., cecum intubation)	12 (75.00%)	34 (64.15%)	16 (66.67%)	0.78
Adenoma detection rate and pathology reports	1 (6.25%)	7 (13.21%)	2 (8.33%)	0.72
Polypectomy technique	4 (26.67%)	22 (41.51%)	13 (54.17%)	0.22
ESGE quality indicator automatically extracted	5 (31.25%)	12 (22.64%)	3 (12.50%)	0.34
Patient’s satisfaction on bowel preparation	0	3 (5.66%)	3 (12.50%)	0.43
Patient’s experience	0	3 (5.66%)	1 (4.17%)	1.00
Complication registry	4 (25.00%)	29 (54.72%)	11 (45.83%)	0.11
Follow-up indication	8 (50.00%)	32 (60.38%)	12 (50%)	0.65

**Table 3 medicina-60-01166-t003:** Structured reporting systems and associated variables at univariate and multivariate analysis.

	OR (95%CI) Univariate	OR (95% CI) Multivariate
Gender female	1.6 (0.72–3.52)	6.22 (2.10–18.41)
Age < 50	0.93 (0.39–2.22)	-
Region North-west North-east South and islands	0.46 (0.12–1.70) 0.89 (0.25–3.10) 0.63 (0.20–1.95)	-
Community versus private hospital	0.87 (0.32–2.38)	-
CCR screening center	1.35 (0.49–3.72)	-
<5-year-old reporting system	4.75 (1.87–12.03)	5.51 (1.96–15.48)
Periodical audit	2.26 (0.91–5.62)	-
Knowledge of quality indicators	0.53 (0.34–0.82)	-

## Data Availability

Data are contained within the article.

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
