# Peer review of "Barriers against Implementation of European Society of Gastrointestinal Endoscopy Performance Measures for Colonoscopy in Clinical Practice"

_medicina, 2024, doi:10.3390/medicina60071166_

Round 1

Reviewer 1 Report

Comments and Suggestions for Authors

Your study indicates that the reporting system for colonoscopy in Italy is not standardized at all. Currently, many countries have their own national registry data base to audit the quality of colonoscopy. If this report can urge your government or related society to regulate the format report that the quality indicator will be accomplished very soon.

Author Response

Comment: Your study indicates that the reporting system for colonoscopy in Italy is not standardized at all. Currently, many countries have their own national registry data base to audit the quality of colonoscopy. If this report can urge your government or related society to regulate the format report that the quality indicator will be accomplished very soon.

Reply: We thank the Reviewer for his suggestion and we added this point in the discussion as follows: “. Other countries are showing how important it is to set up national registers to monitor quality in colonoscopy [23, 24]; in our area, the high heterogeneity and reporting methods make a detailed picture difficult, but certainly the intention of our research was to raise this issue”.

Reviewer 2 Report

Comments and Suggestions for Authors

This is a survey study evaluating the possibility of extracting ESGE performance measures of quality on colonoscopy from endoscopy reporting systems. ESGE has launched a set of recommendations to design reporting systems and it is interesting to know to what extent those recommendations are followed. As expected, the real-life ability of reporting systems to provide these quality indicators is very limited, with only 10.7% of respondents being able to report the adenoma detection rate, for instance. Automatic retrieval of indicators, one of the main ESGE recommendations, was only available in 21.5% of cases.

Main comments:

-       About the questionnaire: 

o   How were the questions formulated? I guess that some of the questions would be of a yes/no design, but others could be formulated using a Likert scale. For instance, the question about the degree of knowledge can be formulated in several ways and it is considered the main aim of the study. Perhaps a copy of the questionnaire could be provided as an addendum. 

o   How were the indicators selected for the survey? The ESGE recommends focusing first on the main indicators to train and prepare the endoscopy unit and the endoscopists to report and retrieve quality indicators, and only after this goal is achieved go for the minor indicators. It seems that key performance measures were selected, but not all. Some minor measures were also selected, but not withdrawal time. 

o   Adenoma detection and information about the resection of lesions belong more to an “intraprocedure” domain than a pre-procedure domain.

o   Did all respondents answer all questions?

o   Was the questionnaire previously tested in a pilot trial?

-       Considering the “degree of knowledge of the ESGE performance domains” as the main aim of the study seems too strong, as it is answered only with one question on the survey.

-       A better description of the outcomes to be measured to fulfill the aims of the study would be advisable. For instance, table 2 seems to summarize the outcomes measured to evaluate the second main outcome. It would be easier for the reader to know that “the actual possibility to systematically assess the ESGE performance measures” will be evaluated by comparing the proportion of extraction of the main performance measures between academic, community, and private centers.  

-       Results:

o   Surprisingly, academic centers perform worse in retrieving quality indicators. Academic hospitals were the less represented in the sample, but this result deserves an explanation in the discussion. 

o   In the multivariable analysis, only a “community hospitals vs private centers” variable was included in the model. Why not an “academic vs non-academic” variable? This would allow us to dig more into differences across centers

-       Discussion:

o   The authors give the high adherence rate as a strength of the study. And it is, indeed, but there were 100 attendants for a quality meeting. We could expect them to be beforehand motivated for quality screening. To what extent do they represent a picture of the situation in Italy? What percentage of the existing endoscopy units do they represent?

o   Some recommendations on how to improve the reporting of quality measures in the Italian setting would be of interest to the reader. 

Minor comments:

-       What is a “periodical audit”? Something like the JAG system in the UK?

-       In Table 2, the academic hospital complication registry lacks the decimals.

Table 2 headline should be more self-explicative.

Author Response

Comment: This is a survey study evaluating the possibility of extracting ESGE performance measures of quality on colonoscopy from endoscopy reporting systems. ESGE has launched a set of recommendations to design reporting systems and it is interesting to know to what extent those recommendations are followed. As expected, the real-life ability of reporting systems to provide these quality indicators is very limited, with only 10.7% of respondents being able to report the adenoma detection rate, for instance. Automatic retrieval of indicators, one of the main ESGE recommendations, was only available in 21.5% of cases.

Main comments:

-       About the questionnaire:

o   How were the questions formulated? I guess that some of the questions would be of a yes/no design, but others could be formulated using a Likert scale. For instance, the question about the degree of knowledge can be formulated in several ways and it is considered the main aim of the study. Perhaps a copy of the questionnaire could be provided as an addendum.

We thank the reviewer for the suggestion. We added more data about the questions’ modality, and we also added a copy of the original questionnaire in the supplementary materials.

o   How were the indicators selected for the survey? The ESGE recommends focusing first on the main indicators to train and prepare the endoscopy unit and the endoscopists to report and retrieve quality indicators, and only after this goal is achieved go for the minor indicators. It seems that key performance measures were selected, but not all. Some minor measures were also selected, but not withdrawal time.

We thank the Reviewer for the comment. As reported in the materials and methods, we selected  the items combining the ESGE performance measures for lower GI endoscopy, ESGE guidelines on bowel preparation and ESGE guidelines on endoscopy reporting systems. Withdrawal time is one of the minor performance measures and we decided to exclude it according to the fact that it depends on withdrawal time technique and it is considered a supportive tool in case of ADR less than the minimum standard of 25 %; based on the fact that many Italian centres already do not promote standardised ADR monitoring, we thought we would focus on this point first as seeing the withdrawal time was of scarce attendance and utility.

o   Adenoma detection and information about the resection of lesions belong more to an “intraprocedure” domain than a pre-procedure domain.

We agree with the Reviewer, we considered a ready pre-procedure reporting system with as many detailed questions as possible so that at the end of the examination the endoscopist is obliged to enter all details. For future formulation, the polypectomy technique and the devices used can be included in the intraprocedure section.

o   Did all respondents answer all questions?

Yes, we confirm that among the 93 completing the survey we observed 100% of respondents.

o   Was the questionnaire previously tested in a pilot trial?

The questionnaire was not tested in a pilot trial; maybe this could be part of further research on this topic.

-       Considering the “degree of knowledge of the ESGE performance domains” as the main aim of the study seems too strong, as it is answered only with one question on the survey.

We thank the Reviewer for this question. We considered the aim of the study explained in two main points: “the degree of ESGE performance measures and the actual possibility to systematically assess and track ESGE performance measures through electronic reporting systems”.

-       A better description of the outcomes to be measured to fulfill the aims of the study would be advisable. For instance, table 2 seems to summarize the outcomes measured to evaluate the second main outcome. It would be easier for the reader to know that “the actual possibility to systematically assess the ESGE performance measures” will be evaluated by comparing the proportion of extraction of the main performance measures between academic, community, and private centers. 

We thank the Reviewer for this comment. As reported in the results and in the table 2 the outcomes were evaluated as percentage according to obtained answers; furthermore we considered a comparison among the proportion of extracted data between the different hospital setting and no significant results were observed.

-       Results:

o   Surprisingly, academic centers perform worse in retrieving quality indicators. Academic hospitals were the less represented in the sample, but this result deserves an explanation in the discussion.

We thank the Reviewer for his comment. We modified the discussion as follows: “Academic hospitals were the least represented in the sample, but this point must be assessed in view of the overlap that is often observed in Italy between community centres and university locations. In other words, we wanted to define whether the presence of a training course has so far led to better investments in terms of technology in reporting systems. Although the training of young endoscopists takes place there on a daily basis, the academic locations are also burdened by the same problems as the other institutes”.

o   In the multivariable analysis, only a “community hospitals vs private centers” variable was included in the model. Why not an “academic vs non-academic” variable? This would allow us to dig more into differences across centers

We thank the Reviewer for the comment. This point can be explained considering that in Italy many community Hopitals are also the location of the most important academic centres and therefore public health and university training are often overlapping environments. Private healthcare centres, on the other hand, are separate facilities where training or activities for residents are not included. We added this explanation in the Discussion.

-       Discussion:

o   The authors give the high adherence rate as a strength of the study. And it is, indeed, but there were 100 attendants for a quality meeting. We could expect them to be beforehand motivated for quality screening. To what extent do they represent a picture of the situation in Italy? What percentage of the existing endoscopy units do they represent?

We thank the Reviewer for the question. As reported in the results: “The majority of participants  was represented by heads of endoscopy units or doctors representing in any case medium or high volume centres”. Figure 1 also shows a good distribution of participants among the national area.

o   Some recommendations on how to improve the reporting of quality measures in the Italian setting would be of interest to the reader.

We thank the reviewer for this suggestion. We added some recommendations in the conclusion paragraph.

Minor comments:

-       What is a “periodical audit”? Something like the JAG system in the UK?

We have no periodical audit on a nationwide scale on this topic; we asked for periodical audit promoted by single regions or hospitals.

-       In Table 2, the academic hospital complication registry lacks the decimals.

We corrected the value.

Table 2 headline should be more self-explicative.

We modified the table 2 headline.

Reviewer 3 Report

Comments and Suggestions for Authors

The study explores the barriers to the implementation of the European Society of Gastrointestinal Endoscopy (ESGE) performance measures for colonoscopy in clinical practice in Italy. This article is commendable in terms of novelty, scientific rigor, professionalism, methodological soundness, comprehensiveness of results presentation, and clarity of language. The research design is well-constructed, data analysis methods are scientific, results are comprehensively presented, and the language is clear and concise, making this a study of high academic value and practical significance. However, there are several areas for improvement:

1.  While the sample includes multiple centers across Italy, the sample size is relatively small (93 participants). Increasing the sample size would enhance the generalizability and robustness of the findings.

2. The article analyzes the results from different types of centers (academic hospitals, community hospitals, private hospitals), but further exploration of the reasons behind these differences is needed. For instance, what specific barriers do different centers face in implementing ESGE performance measures? Are there unique management models or policies that contribute to these differences?

3. In the discussion section, incorporating a comparative analysis with similar studies from other countries would highlight the unique contributions and significance of this study. For example, examining the successful experiences and implementation strategies of other countries, particularly those that have achieved efficient data extraction and performance tracking systems, would provide valuable insights.

4.The conclusion section could more explicitly state the limitations of the study and propose directions for future research. Suggestions could include conducting further multicenter and large-sample studies, exploring new reporting systems and technologies, and comparing the implementation outcomes in different countries.

Author Response

Comment: The study explores the barriers to the implementation of the European Society of Gastrointestinal Endoscopy (ESGE) performance measures for colonoscopy in clinical practice in Italy. This article is commendable in terms of novelty, scientific rigor, professionalism, methodological soundness, comprehensiveness of results presentation, and clarity of language. The research design is well-constructed, data analysis methods are scientific, results are comprehensively presented, and the language is clear and concise, making this a study of high academic value and practical significance. However, there are several areas for improvement:

  1. While the sample includes multiple centers across Italy, the sample size is relatively small (93 participants). Increasing the sample size would enhance the generalizability and robustness of the findings.

We agree that a larger sample size would enhance the robustness of our research. However, the survey was conducted during during a Nationwide Event on quality of colonoscopy. All participants were gastroenterologists performing endoscopy and more precisely many of them were Head of Department of Chiefs of Endoscopy Units so we think that they were quite representative for raising the issue of heterogeneous reporting systems in Italy.

  1. The article analyzes the results from different types of centers (academic hospitals, community hospitals, private hospitals), but further exploration of the reasons behind these differences is needed. For instance, what specific barriers do different centers face in implementing ESGE performance measures? Are there unique management models or policies that contribute to these differences?

We thank the Reviewer for the question. The specific barriers present today are first of all related to the lack of availability of a single electronic reporting system on a national scale, but even before that on a regional scale and also in the city itself. Each institute has its own regulations. National societies (e.g. SIED) promote homogeneity in the performance of quality endoscopy, but surely the intention of this survey is to provide details on which to focus and work in the coming years.

  1. In the discussion section, incorporating a comparative analysis with similar studies from other countries would highlight the unique contributions and significance of this study. For example, examining the successful experiences and implementation strategies of other countries, particularly those that have achieved efficient data extraction and performance tracking systems, would provide valuable insights.

We thank the Reviewer for his comment. We had already cited the experience of other countries in Europe and outside Europe adopting standardized and comparable reporting systems [20-22]. We added also a comment about the possibility to use these tools to improve the quality in colonoscopy offered by the Hospital not focusing on the single endoscopist performance, as follows “The use of established systems that allow for data extraction comparable among different institutions offers several advantages already shown in recent literature. Belderbos et al. showed their experience in 2015 by comparing quality parameters of routine colonoscopies between two academic and five nonacademic hospitals in The Netherlands. They included 3129 consecutive patients undergoing colonoscopy and they proposed a colonoscopy quality indicator (CQI) by combining cecal intubation rate (CIR) and ADR [23]. This new tool should lead to an evaluation that is less focused on the performance of the individual endoscopist and rather promotes changes that benefit the entire institution [24]”.

4.The conclusion section could more explicitly state the limitations of the study and propose directions for future research. Suggestions could include conducting further multicenter and large-sample studies, exploring new reporting systems and technologies, and comparing the implementation outcomes in different countries.

We agree with the Reviewer and we added a conclusive comment as follows “Multicentre and large-sample studies are needed to demonstrate the advantage of a uniform reporting system throughout our country; subsequently, an equal comparison between European countries would allow for measurable progress and a clearer application of guidelines recommendations”.

Round 2

Reviewer 2 Report

Comments and Suggestions for Authors

The authors have responded adequately to the doubts raised by the reviewers. Congratulations.